# A Simplified Relationship Between the Zero-percolation Threshold and Fracture Set Properties

Shaoqun Dong<sup>1,2,3\*</sup>, Lianbo Zeng<sup>1,4</sup>, Chaoshui Xu<sup>3</sup>, Peter Dowd<sup>3</sup>, Guohao Xiong<sup>1,2</sup>, Tao Wang<sup>5</sup>, Wenya Lyu<sup>1,4</sup>

- <sup>1</sup> National Key Laboratory of Petroleum Resources and Engineering, China University of Petroleum (Beijing), Beijing, China. 102249
  - <sup>2</sup> College of Science, China University of Petroleum, Beijing, China 102249
  - <sup>3</sup> School of Civil, Environmental and Mining Engineering, University of Adelaide, Adelaide, Australia 5005
  - <sup>4</sup> College of Geosciences, China University of Petroleum, Beijing, China 102249
- <sup>5</sup> Chinese Academy of Geological Sciences, Beijing, China 100037

Correspondence to: Shaoqun Dong (dshaoqun@163.com) ORCID: 0000-0001-8204-7336

Abstract. Percolation analysis is an efficient way of evaluating the connectivity of discrete fracture networks. Except for very simple cases, it is not feasible to use analytical approaches to find the percolation threshold of a discrete fracture network. The most commonly used percolation threshold corresponds to the occurrence of percolation on average for the set of parameters (p50), which is not adequate for applications in which a high confidence in the percolation threshold is required. This study investigates the direct relationships between the percolation threshold at low probability (p0, referred to as zero-percolation threshold) and the properties of fracture networks with one set of fractures (fractures with similar orientations) in two-demensional domains. A generalized non-linear multivariate relationship between p0 and fracture network parameters is established based on connectivity assessments of a significant number of numerical simulations of fracture networks. A feature of this relationship is the invariant shape of marginal relationships. A comparison study with an analytical solution and applications in both synthetic and real fracture networks shows that the derived relationship performs well in fracture networks of different sizes and orientations. A significant benefit of this relationship is that, when an analytical solution is not available, it can provide fast and reliable connectivity statistics of fracture networks based only on fracture parameters.

**Keywords**: Percolation; percolation threshold; fracture network; connectivity; discrete fracture network.

#### 1 Introduction


Discrete fracture networks (DFN) are widely used to model fracture systems in rock masses and reservoirs for flow analysis (Dogan, 2023; Kolyukhin, 2022; Liu et al., 2019). In three-dimensional DFNs, fractures are represented using simplified

geometric shapes such as polygons, rectangles, disks, and others. In two-dimensional DFNs, fractures are simplified as line segments (Dong et al., 2020). DFN models are generally considered to be more realistic representations of the fracture systems and they are more amenable to integrating geological data into the flow model (Dong et al., 2018). Fracture connectivity analysis is a significant component in the DFN approach to assessing flow behaviour (Alghalandis et al., 2015; Einstein and Locsin, 2012) as the conductivities of fractures are generally orders of magnitude greater than those of the surrounding porous matrix (Thovert et al., 2017). A good understanding of the connectivity of fracture networks is essential for many applications such as oil and gas recovery, geothermal energy exploitation, hydrology and groundwater engineering, and geological storage of radioactive wastes.



Percolation theory (Jafari and Babadagli, 2013; Khamforoush and Shams, 2007; Or et al., 2023; Sun et al., 2023) provides a basis for describing and quantifying the connectivity of geometrically complex systems (Xu et al., 2007) such as fracture networks and this is reflected in many studies reported in the literature that use percolation theory in the connectivity analysis of fracture systems (Barker, 2018; de Dreuzy et al., 2000; Dong et al., 2022; Khamforoush and Shams, 2007; Manzocchi, 2002; Masihi and King, 2007; Mourzenko et al., 2012). Percolation describes the phenomenon in which there is at least one domain-spanning pathway in a physical system (Tang et al., 2022; Yao et al., 2020; Yi and Tawerghi, 2009). Percolation in a DFN involves at least one cluster of connected fractures that spans the reservoir (McKenna et al., 2020) or rock mass (one cluster refers to a series of fractures that intersect each other). In this context, one of the most important characteristics of a fracture network is whether or not it percolates (Bour and Davy, 1998; Bour and Davy, 1997) and the percolation threshold is commonly used to quantify the critical value of connectivity at which the network percolates (Khamforoush and Shams, 2007; Manzocchi et al., 2023; Walsh and Manzocchi, 2021). Using this definition, the permeability of a fracture network is zero if the connectivity value is less than the percolation threshold (Mourzenko et al., 2005).

The percolation threshold of a DFN is a property that depends on the parameters of the fracture system (Mourzenko et al., 2005). Features previously used to characterise percolation in DFNs include the dimensionless density derived from the excluded volume (Barker, 2018; de Dreuzy et al., 2000; Khamforoush et al., 2008; Mourzenko et al., 2012), fractal dimensions (Jafari and Babadagli, 2013; Jafari and Babadagli, 2009; Zhao et al., 2016), topological connectivity measures (Manzocchi, 2002), fracture clustering (Manzocchi, 2002), and the average number of intersections per fracture (Manzocchi, 2002). These indirect characteristics of DFN models are derived from direct fracture network parameters, such as the number of fractures, fracture locations, sizes, and orientations, which have a joint effect on the occurrence of a percolating network (Jafari and Babadagli, 2009). For example, if the fracture size is kept constant, an increase in the number of fractures will result in a higher fracture density, which in turn will increase the probability of a connected domain (Shokri et al., 2016).

There are many published studies on the percolation of DFN models, with different focuses on different aspects of the problem. For fracture locations, DFN models in these studies cover both the Poisson (homogeneous) distribution (Barker, 2018; Bour and Davy, 1997; de Dreuzy et al., 2000; Huseby and Thovert, 1997; Jafari and Babadagli, 2013; Khamforoush and Shams, 2007; Mourzenko et al., 2005; Robinson, 1983; Thovert et al., 2017; Zhao et al., 2009; Zhao et al., 2016) and non-homogeneous (i.e., spatially correlated) distributions (Manzocchi, 2002; Mourzenko et al., 2012). For fracture sizes, some

DFNs use the monodisperse model, which means that the shape and size of every fracture are identical (Jafari and Babadagli, 2013; Khamforoush and Shams, 2007; Khamforoush et al., 2008; Manzocchi, 2002; Mourzenko et al., 2012; Robinson, 1983). This makes the percolation study relatively simple using numerical simulation. Others use the polydisperse model in which the sizes (Bour and Davy, 1997; Huseby and Thovert, 1997; Mourzenko et al., 2004; Mourzenko et al., 2005; Thovert et al., 2017; Zhao et al., 2016) and shapes (Barker, 2018; Thovert et al., 2017) of fractures are different. Commonly used fracture size distributions include the power law function (Mourzenko et al., 2004; Mourzenko et al., 2005; Zhao et al., 2016), exponential distribution (Catapano et al., 2023; Dowd et al., 2007; Fadakar Alghalandis, 2017; Xu et al., 2007; Zhu et al., 2022) and uniform distribution (Huseby and Thovert, 1997). For fracture orientations, many DFNs use isotropic models (uniform and random) (Barker, 2018; Bour and Davy, 1997; Charlaix et al., 1984; de Dreuzy et al., 2000; Huseby and Thovert, 1997; Jafari and Babadagli, 2013; Khamforoush and Shams, 2007; Khamforoush et al., 2008; Mourzenko et al., 2004; Mourzenko et al., 2012; Mourzenko et al., 2005; Robinson, 1983; Thovert et al., 2017; Yi and Tawerghi, 2009; Zhao et al., 2016) but anisotropic (with a single preferential orientation or several preferential orientations) models are also commonly used (Balberg and Binenbaum, 1983; Khamforoush and Shams, 2007; Khamforoush et al., 2008; Manzocchi, 2002). The Fisher distribution (Khamforoush and Shams, 2007; Xu and Dowd, 2010) is the most commonly used type of distribution for three-dimensional fracture networks, while the von Mises distribution (Xu and Dowd, 2010) is the most commonly used for two-dimensional fracture networks. Physically, fractures related to tectonic movements are, in general, anisotropic (e.g., conjugate fractures generated around the maximum principal compressive stress (Zhao and Hou, 2017), while fractures associated with other causes, such as diagenesis, are typically isotropic (Dong et al., 2018).




A common method used to obtain the percolation threshold of a DFN is first to calculate some indirect characteristic parameters of the fracture network and then evaluate the percolation threshold on the basis of these parameters. However, DFNs with the same indirect characteristic parameters may have quite different direct geometrical parameters (e.g., number of fractures, fracture size, and orientation). Unfortunately, these geometrical parameters dictate the fracture connectivity and hence the percolation threshold (Dong et al., 2019). For example, the two fracture networks in Figure 1 have the same number of fractures, identical fracture lengths, and box-counting fractal dimensions, but the network in Figure 1a percolates between sides A and B, while the other (Figure 1b) does not percolate. The different orientations of these two fracture models result in different percolation characteristics. In this case, the box-counting fractal dimension provides a good measure of the complexity of the system but it ignores the effect of the preferential orientation of a fracture network. Although the indirect approach can simplify the evaluation of the percolation threshold of a fracture network, it may sometimes produce misleading results. In addition, most percolation thresholds based on the excluded volume method correspond to the occurrence of percolation on average (Barker, 2018; Yi and Tawerghi, 2009) (i.e., with 50% probability, p50) due to the stochastic nature of fracture networks (c.f. Section 2.2). The level of confidence in such thresholds may not be sufficient for some applications. For example, for underground radioactive waste storage facilities, selecting storage sites that minimize the potential number of connected pathways to the biosphere is important. In this case, a low probability percolation threshold (p0, c.f., Section 2.2) named as zero-percolation threshold is more important for the connectivity analysis of the fracture systems (Dong et al., 2019).

Figure 1: Schematic diagram of two DFN models with different percolation features.






In practical engineering applications, it would be less problematic to use the direct relationship between the percolation threshold and fracture network parameters for connectivity assessments instead of resolving the problem by numerical simulations in every case. In general, it is not possible to establish such a relationship analytically due to the complexity of DFNs. In this work, Monte Carlo simulations of the number of fractures, n, fracture size (length), L, and fracture orientation,  $\phi$  are used to establish this relationship for two-dimensional DFN models, the fractures are represented by vectors representing line segments (Dong et al., 2020). In particular, fracture locations follow the Poisson distribution, fracture lengths follow the exponential distribution  $f(L|\lambda)$  ( $\lambda$ : the exponential distribution parameter whose reciprocal  $1/\lambda$  represents the mean fracture length  $\bar{L}$ ) and fracture orientations follow the von-Mises distribution  $f(\phi|\mu,\kappa)$ , where  $\lambda$ ,  $\mu$  and  $\kappa$  are their corresponding distribution parameters,  $\mu$  represents the mean/main fracture orientation, while  $\kappa$  (concentration parameter) quantifies the directional clustering degree of fractures in the von Mises distribution.". (see Section 2.1). The zero-percolation threshold (p0) equation  $L_t = f(n, \mu, \kappa)$  was established by analysing results from an extensive set of numerical simulations for fracture networks with one set of fractures (fractures that exhibit similar orientations) (Ali and Jakobsen, 2011; Zeng et al., 2022) (see Sections 3.1-3.2). Besides DFN with exponential fracture length, given the widespread adoption of lognormal distribution  $f(L|\mu,\sigma)$  in characterizing fracture length distributions within DFN, it is imperative to explore the implications of this distribution on the phenomenon of zero-percolation. Here,  $\mu$  and  $\sigma$  are parameters of the lognormal distribution, where  $\mu$  is the mean of the natural logarithm of fracture lengths and  $\sigma$  is the standard deviation of the natural logarithm of fracture lengths (derived from the mean and standard deviation of fracture lengths). Consequently, this paper extends its investigation beyond DFNs employing exponential distribution for fracture lengths to encompass those utilizing lognormal distribution, as detailed in Sections 3.3-3.4.

The relationship was established by using a non-linear, multivariate fitting method for a relationship with invariant shapes of marginal functions; this is demonstrated by a simple example in Sections 2.2 and 2.3. The verification of the derived equations for zero-percolation underwent a comprehensive series of tests in Section 4.

## 2 Principle of the mathematical method

To mitigate ambiguity, the mathematical approach will utilise DFN with exponential distributions as an instance to elucidate the fundamental principles. Fracture parameters of a DFN model are described in Section 2.1. The percolation and percolation threshold of a DFN are described in Section 2.2. In Section 2.3, the non-linear fitting method is illustrated using a simple percolation example.

#### 2.1 Discrete fracture networks

DFN modelling is a stochastic simulation method that uses marked point processes (MPP) (Dong et al., 2018c) in which fracture location (x, y) is modelled by a point process (Figure 2a) following a Poisson, non-homogeneous cluster or Cox point process (Mardia et al., 2007); and fracture properties (such as length L, orientation φ) are modelled at each point by marks (Figure 2b) following their respective probability distribution functions (e.g., f(L) and f(φ)) (Dong et al., 2018; Fadakar Alghalandis, 2017; Xu and Dowd, 2010). To simulate a set of n fractures with similar orientations, the location of a fracture
(Figure 2a) is generated first followed by the generation of the associated marks (Figure 2b). Subsequently, these procedures are iterated n times to culminate in the ultimate implementation of the DFN (Figure 2c).

Figure 2: Schematic diagram of a two-dimensional DFN realization. (a) Randomly generated fracture location; (b) Fracture properties (length, orientation) are generated from their probability distributions; (c) Repeat process (a) and (b) to generate the entire fracture network to account for the number of fractures in each fracture set as well as the number of fracture sets.

For 2D DFN models, fracture network parameters include the number of fracture sets, number of fractures n in each fracture set, fracture size distribution f(L) and fracture orientation distribution  $f(\phi)$ . The study area used in this work is  $100m \times 100m$  so  $P20 = n_i \times 10^{-4} (m^{-2})$ . P20 (2D fracture density) represents the number of fractures per unit area. Here, P20 is the fracture number per 2D unit area (Khamforoush et al., 2008). For fracture length, a fixed size L can be used, or the following exponential distribution (Xu and Dowd, 2010) is commonly used:

$$f(L|\lambda) = \begin{cases} \lambda e^{-\lambda L} & L \ge 0\\ 0 & L 

Figure 3. DFN models showing different fracture orientations with different  $\mu$ ,  $\kappa$ , together with their corresponding rose diagrams. Fractures of the same colour in the DFN model are in the same cluster, while fractures in black are isolated ones.

## 2.2 Percolation threshold of DFN models

Percolation in a DFN means there is at least one cluster of fractures spanning the system (rock mass or reservoir) that allows the fluid to permeate from one side to the other, as shown in Figure 4b, while Figure 4a shows a non-percolating DFN. Obviously, one can easily conclude that an increase in fracture number or fracture size can lead to the percolation of the system at some stage. In reality, the probability of percolation is a function of many different factors related to the DFN parameters (Khamforoush et al., 2008), of which fracture density (such as P20 and P21 for 2D and P30 and P32 for 3D applications) is the most critical. Here, P21 is the fracture length per 2D unit area, while P30 and P32 are the fracture number and area per 3D unit volume, respectively.

Figure 4: Schematic diagram of percolation in DFN models.





For simple fracture networks, the percolation thresholds can be found analytically. However, for most fracture networks, approaches such as Monte Carlo (MC) simulation are required (Yi and Tawerghi, 2009). Due to the random nature of a DFN model, its corresponding percolation status is also stochastic in nature. If N independent simulations of a DFN are repeated for a group of parameters  $(n, L, \phi)$ , resulting in  $N_p$  number of cases where the fracture network percolates, then the percolation probability corresponding to the parameter set is  $P = N_p/N$  (Barker, 2018; Yi and Tawerghi, 2009). In this study, p0, p50, and p100 represent the percolation thresholds of the DFN corresponding to 0%, 50%, and 100% percolation, respectively. p0 (zero-percolation threshold) is defined as the critical value of a fracture parameter below which network connectivity is completely lost, ensuring absolute impermeability. In a fracture network with two variables, fracture number (n) and fracture length (L), p0 may be represented either by a sufficiently small n (regardless of L) or by a minimum L when n is fixed.

As illustrated in Figure 5, percolation probability typically transitions between P=0 (no percolation) and P=1 (full percolation) through an intermediate band. The choice of percolation threshold depends on the application. For example, in subterranean radioactive waste repositories, where preventing any potential connectivity to aquifers is critical (Wei et al., 2017; Yi and Tawerghi, 2009), p0 is adopted to ensure absolute impermeability. Conversely, for applications such as unconventional gas extraction or enhanced geothermal systems, near-complete connectivity is desired, and the threshold may correspond to P=1 p50 is often adopted in resource extraction (e.g., hydrocarbon reservoirs) to balance economic viability and manageable

risk. Noted in stochastic systems, P = 0% and 100% may not be strictly possible and therefore the definition used here means the probability calculated by  $N_p/N_T$  using a reasonable number  $N_T$  (= 20 in this study). For stochastic systems, Figure 5 shows a worked example, in which the fracture orientations follow a von-Mises distribution ( $\mu = 90^\circ$ ,  $\kappa = 24$ ) and fracture lengths are identical for each DFN. Twenty MC simulations ( $N_T = 20$ ) were conducted using pairs of parameters (n = 20,30,40,...,250 and L = 0.06,0.08,0.1,...0.8). The percolation probability, P, calculated from the simulations, is shown in Figure 5, where the horizontal axes correspond to the number of fractures n and the fracture length L, respectively, and the z axis is the percolation probability of the corresponding DFN model.

Figure 5: Percolation probability P versus fracture number n and fracture length L for DFN models with random orientations and identical lengths. Each P value is obtained from 20 Monte Carlo simulations for a given (n,L) pair. For each n, the fracture length threshold corresponding to the p0 zero-percolation point is determined, and the average over 20 repeats is taken as the statistical p0 threshold.

Figure 6a presents data points in the vicinity of the p0 percolation threshold. For each fracture number (n), the statistical mean of the fracture length threshold  $(L_t)$  corresponding to p0 percolation is determined from 20 independent Monte Carlo (MC) simulations, following the same procedure as in Figure 5. The mean values are depicted as circles, with half error bars indicating the mean  $\pm$  two standard deviations in Figure 6a. To further approach the true threshold values, an additional 20 repetitions of the Figure 6a procedure are performed, and the results are shown in Figure 6b. In the subsequent analyses, a total of 400 MC simulations [20 (as in Figure 6a)  $\times$  20 (as in Figure 6b)] are conducted for each data point, providing a statistically robust estimate that closely approximates the true value. In order to reduce the computation cost, there are a few differences in fracture number n and length L compared with those in Figure 5, i.e.,  $n = round(10^j) = 22, 32, 45, ..., 251, j = 1.35, 1.5, 1.65, ..., 2.4, <math>L = 10^k = 0.0316, 0.0322, 0.0327, ..., 1, k = -1.5, -1.4925, -1.485, ..., 0$ . The corresponding non-linear percolation threshold curve based on least-squares regression is  $L_t = 1.7518 \times n^{-0.4308}$  with a correlation coefficient of 0.9288. When the specific number n is considered, an increase in L results in percolation. In this context, we hold the fracture number n to determine the fracture length threshold (denoted as  $L_t$ ) for zero probability percolation, hence the utilisation of  $L_t$ 

and n. Conversely, if the fracture length is fixed, L and  $n_t$  will be employed. This curve defines the percolation threshold in terms of parameter pairs of  $(n, L_t)$ . The DFN corresponding to any combination of parameters below this curve, namely  $L 


(a)  $L_t$  vs. n from 20 MC simulations in Figure 5, with circles showing means and half error bars indicating twice the standard deviation.

(b)  $L_t$  vs. n. Each black point represents an average corresponding to a red circle in (a). For each n, (a) is repeated 20 times.

(c)  $(n, L_t)$  corresponding to different  $\Delta$ .

Figure 6: Percolation threshold of  $(n, L_t)$  for the example DFN model.

Obviously, fracture orientations will also affect the percolation threshold curves. As the example to demonstrate these effects, the von-Mises distribution is used to describe the distribution of fracture orientations for the DFN model used above. In this case, the fracture network parameters are  $(n, L, \mu, \kappa)$ . To simplify the demonstration, the concentration parameter  $\kappa$  is set to 24, similar to those shown in Figure 3b and Figure 3d;  $\mu$  is set to values from 90° to 0° in 5° decrements. As the horizontal percolation is of interest here, to simplify the comparison,  $\mu$  is transformed to an angle measured from the horizontal direction, i.e.,  $\Delta = |\mu - 90|$  (Relative orientation angle, indicating the deviation angle between the dominant fracture direction  $\mu$  and the horizontal direction, used for simplified horizontal percolation analysis), hence,  $\Delta = 0^{\circ}$ , 5°, ..., 90°. The above curve fitting process was repeated and some results are shown in Figure 6c. As  $\Delta$  decreases, the percolation threshold decreases. This is consistent with the fact that lower  $\Delta$  will increase the connectivity in the horizontal direction between the left and the right sides; percolation therefore requires a shorter fracture length, and so the percolation threshold curve decreases.

## 2.3 Non-linear relationship for the percolation threshold





The general shapes of the percolation curves in Figure 6c can be considered to have similar shapes, which are represented as coloured curves in Figure 7. To understand further the relationship between  $L_t$ , n and  $\Delta$ , average values of 20 MC simulations are shown in Figure 7. Coloured curves refer to slices of different n and  $\Delta$ . The left colour curves are the variation of  $L_t$  and n with different values of  $\Delta$ , and the right coloured curves are the variations of  $L_t$  and  $\Delta$  with different values of n. Clearly, the threshold fracture length increases as the number of fractures decreases and the relative orientation  $\Delta$  increases. This is because higher fracture density leads to greater percolation probability, whereas a lower number of fractures requires longer fractures to maintain the same percolation probability.

Figure 7: Simulated percolation threshold  $L_t$  vs.  $(n, \Delta)$ . For each pair of  $(n, \Delta)$ , 20 MC simulations are implemented to obtained the average  $L_t$ . Coloured curves are lines corresponding to slices of different n and  $\Delta$ .

From the results in Figure 7,  $L_t = f(n, \Delta)$  is non-linear. To establish this relationship, the variation of  $L_t$  with n for different values of  $\Delta$  is examined first, i.e.,  $L_t = f_1(n)|_{\Delta}$ , followed by assessing the influence of  $\Delta$  on the derived  $f_1(n)$  relationship. Note that at the second stage,  $\cos \Delta$  is used instead of  $\Delta$  as it is more relevant to the quantification of the fracture projection length in the horizontal direction ( $len/\cos \Delta$ ).

Based on the simulation results discussed above, Eq. 3 is considered an appropriate fit to  $f_1(n)$ :

$$L_t = f_1(n) = an^b, (3)$$

where a and b are parameters to be determined in the fitting process. The correlation coefficients for all curves for different  $\Delta$  values (0°,5°...85°, 90°) are 0.9985, 0.9895, ..., 0.9949, respectively. The high correlation coefficients (>0.96) confirm the suitability of using Eq. 3 to represent  $f_1(n)$ .

The relationships between a, b and  $\cos \Delta$ , shown in Figure 8, suggest linear relationships. Least squares regression was used to obtain the following equations:

$$a = a_1 \cos^2 \Delta + a_2 \cos \Delta + a_3, \tag{4}$$

$$b = b_1 \cos^2 \Delta + b_2 \cos \Delta + b_3, \tag{5}$$

with correlation coefficients of 0.9919 and 0.9712, respectively;  $a_1$ ,  $a_2$ ,  $a_3$ ,  $b_1$ ,  $b_2$ ,  $b_3$  are 2.726, -5.304, 2.887, -0.2552, 0.6134, and -0.5724, respectively.

Figure 8: Influence of  $\Delta$  on a and b.




By incorporating Eqs. 4 and 5 into Eq. 3 the final form of the expression of  $L_t$  in terms of the fracture network parameters is obtained, as shown in Eq. 6. This form is then used directly in a bivariate least squares fitting using the Levenberg-Marquardt algorithm (Ngia and Sjoberg, 2000), an optimal search technique for multivariate non-linear curve fitting. The

original values of parameters shown in Eqs. 4 and 5 are used as initial inputs to the optimisation algorithm to improve computational efficiency and accuracy and the final derived parameters in this case are  $(a_1, a_2, a_3, b_1, b_2, b_3) =$ (2.4757, -4.9064, 2.7359, -0.1841, 0.5097, -0.5336). This set of values should be a more accurate reflection of the bivariate relationship than the values obtained in the two separate consecutive steps described above. The  $\bar{L}_t$  parameter (mean fracture length threshold) quantifies the orientation-dependent critical length for percolation, defined mathematically as:

$$\bar{L}_t = (a_1 \cos^2 \Delta + a_2 \cos \Delta + a_3) n^{(b_1 \cos^2 \Delta + b_2 \cos \Delta + b_3)}, \tag{6}$$

The final fitted surface is shown in Figure 9a. The points are the average values of 20 groups of MC simulation results shown in Figure 7. The suitability of the chosen functional form (Eq. 6) is confirmed by the fact that almost all the points are on the fitted surface. The plot in Figure 9b of simulated values of  $L_t$  against those predicted by Eq. 6 gives an extremely high correlation coefficient of nearly 1. It is also encouraging that, on visual inspection, the fitted curve is conditionally unbiased. Although this workflow is useful for multivariate non-linear fitting problems in which marginal relationships are of invariant shape, it should be noted that difficulties may arise for very high dimensions (Dong et al., 2016).

The process described above can be summarised as an approach for fitting multiple variables. This approach starts by fitting a hypothetical relationship between  $L_t$  and n is initially fitted. Then, a new variable  $\Delta$  is added by analysing relationships with the parameters in the hypothetical relationship model. The parameters in the hypothetical relationship model are then replaced by expressions of the newly added variable. Ultimately, the relationship between between  $L_t$  and  $(n, \Delta)$  can be obtained. This approach will be applied in the relationship fitting of Section 3.2, where the independent variables are  $(n, cos \Delta, \kappa)$ .

100

80

60

40

20






calculated by fitting equation 0 100 L, obtained by simulation (b) Cross plot between simulated values of  $L_t$  and those predicted by Eq. 6

r=0.996

(a) Fitted percolation threshold surface and simulation data

Figure 9: Bivariate percolation threshold fitting.

#### 3 Percolation analysis of DFN models

#### 3.1 Experiment design for DFN with exponential fracture lengths

In the example used above, the lengths and orientations were identical for all fractures in a fracture network, which was not generally the case in practical applications. The relationship described above could be made more useful by extending it to cover realistic fracture networks. The following numerical experiments were all implemented on a dimensionless unit square (1×1). Based on previously published work (Dong et al., 2018; Xu et al., 2007), the lengths of rock fractures could generally be modelled by an exponential or lognormal distribution. In this work, the exponential distribution was used, and therefore the average length  $\bar{L}$  is equal to  $1/\lambda$  where  $\lambda$  was the distribution parameter. The von-Mises distribution (Eq. 2) was used for fracture orientation.

There are now three independent variables  $(n, \Delta, \kappa)$  and the aim is to establish the relationship  $\bar{L}_t = f(n, \Delta, \kappa)$ . To simulate percolation states similar to those shown in Figure 5, DFNs corresponding to a combination of  $8 \times 18 \times 13 \times 200$  (374,400) sets of variables were simulated and analysed, with each case simulated independently 20 times. The number of changes explored for each variable are listed in Table 1.

Table 1: Parameters of DFNs in Section 3.2.





| Parameter             | Values analyzed                                 | Number of values |
|-----------------------|-------------------------------------------------|------------------|
| n                     | $round(10^{i}), i = 1.5, 1.65, 1.8, \dots, 2.4$ | 8                |
| $\Delta =  \mu - 90 $ | $(i-1) \times 5^{\circ}, i = 1,2,3,,18$         | 18               |
| κ                     | $round(10^i), i = 0.602, \dots, 2$              | 13               |
| $\bar{L}=1/\lambda$   | $10^{i}, i = -1.5, -1.4925, -1.485, \dots, 0$   | 200              |

For each pair of  $(\Delta, \kappa)$ , 20 independent realisations of DFNs with different n and  $\bar{L}$  were generated to obtain the percolation threshold curves  $\bar{L}_t = f_1(n)$ . These 20 MC simulations are used to calculate the percolation probability to obtain the points  $(n, \bar{L}_t)$ , as shown in Figure 10. The points are the average values of 20 groups of 20 MC simulations and the error bars represent two times of the corresponding standard deviation. The relationship in Eq. 3 was used again for  $\bar{L}_t = f_1(n)|_{\Delta,\kappa}$ . A comparison of Figure 6b and Figure 10 indicates that the standard deviations in this case are much larger, which is expected due to the variability in the lengths of fractures generated in simulations. Note that the uncertainty (reflected by the size of the error bar) increases as the number of fractures, n, decreases.



Figure 10: Variation of the percolation threshold as a function of n for  $\kappa = 3$  and  $\Delta = 0$ .

## 3.2. Determining percolation threshold equation for DFN with exponential fracture length

Eq. 6 is only for a specific parameter  $\kappa$ . Establishing the full relationship,  $\bar{L}_t = f(n, \cos\Delta, \kappa)$ , requires the relationships between  $(a_1, a_2, a_3, b_1, b_2, b_3)$  in Eq. 6 and  $\kappa$  for different DFNs. The regression results for these parameters for different  $\kappa$  values are shown in Table 2, which provides the input data for the final non-linear fitting of the percolation threshold function. The correlation coefficients for each of the regressions in the table are all greater than 0.98, which ensures the suitability of the derived relationships.

Table 2: Regression parameters  $(a_1, a_2, a_3, b_1, b_2, b_3)$  for different values of  $\kappa$ .

| κ  | a <sub>1</sub> | $a_2$   | $a_3$  | $b_1$    | $b_2$   | $b_3$   |
|----|----------------|---------|--------|----------|---------|---------|
| 3  | 30.1061        | -0.3339 | 0.6003 | -0.03176 | 0.08575 | -0.329  |
| 6  | 0.3431         | -0.8712 | 0.8535 | -0.08643 | 0.2164  | -0.3734 |
| 10 | 0.8132         | -1.835  | 1.338  | -0.1105  | 0.3005  | -0.4269 |
| 13 | 1.328          | -2.77   | 1.765  | -0.1555  | 0.3679  | -0.4494 |
| 18 | 1.883          | -3.781  | 2.199  | -0.2105  | 0.5097  | -0.5119 |
| 24 | 2.726          | -5.304  | 2.887  | -0.2552  | 0.6134  | -0.5724 |
| 32 | 3.212          | -6.382  | 3.45   | -0.3276  | 0.8044  | -0.6671 |
| 56 | 6.14           | -11.59  | 5.724  | -0.2222  | 0.6696  | -0.6211 |
| 75 | 8.671          | -15.94  | 7.582  | 0.04139  | 0.3522  | -0.5445 |

The relationships between  $a_1$  and  $\kappa$ , between  $a_2$  and  $\kappa$  and between  $a_3$  and  $\kappa$  are linear as described by Eqs. 7-9; the relationship between  $b_1$  and  $\kappa$ , between  $b_2$  and  $\kappa$  and between  $b_3$  and  $\kappa$  are quadratic as shown in Eqs. 10-12. The correlation

coefficients of this set of regression curves are all greater than 0.98. Overall, these variables display a clear and strong relationship that can be described by an appropriate functional form. Table 3 lists the constants in Eqs. 7-12 obtained by least squares regression.

Figure 11: Relationship between  $\kappa$  and fitted parameters  $(a_1, a_2, a_3, b_1, b_2, b_3)$ .

$$a_1 = a_{11}\kappa + a_{12},\tag{7}$$

$$a_2 = a_{21}\kappa + a_{22},\tag{8}$$

$$a_3 = a_{31}\kappa + a_{32},\tag{9}$$

$$b_1 = b_{11}\kappa + b_{12}\kappa^2 + b_{13}, \tag{10}$$

$$b_2 = b_{21}\kappa + b_{22}\kappa^2 + b_{23}, (11)$$

$$b_3 = b_{31}\kappa + b_{32}\kappa^2 + b_{33},\tag{12}$$

where  $a_{11}, a_{12}, a_{21}, a_{22}, a_{31}, a_{32}, b_{11}, b_{12}, b_{13}, b_{21}, b_{22}, b_{23}, b_{31}, b_{32}, b_{33}$  are parameters.

Table 3: Parameters of Eqs. (7) - (12).




| a <sub>11</sub> | $a_{12}$ | $a_{21}$ | $a_{22}$ | a <sub>31</sub> | $a_{32}$ | $b_{11}$ | $b_{12}$ |
|-----------------|----------|----------|----------|-----------------|----------|----------|----------|
| 0.1177          | -0.296   | -0.2143  | 0.2192   | 0.096           | 0.4064   | 0.0002   | -0.0182  |
| $b_{13}$        | $b_{21}$ | $b_{22}$ | $b_{23}$ | $b_{31}$        | $b_{32}$ | $b_{33}$ |          |
| 0.0299          | -0.0004  | 0.0377   | -0.0204  | 0.0002          | -0.0165  | -0.279   |          |

Finally, the combined percolation equation  $\bar{L}_t = f(n, \cos \Delta, \kappa)$  can be obtained, as shown in Eq. 13. The correlation coefficient is again nearly 1 (0.99).

$$\bar{L}_{t} = \left( (a_{11}\kappa + a_{12})x^{2} + (a_{21}\kappa + a_{22})x + (a_{31}\kappa + a_{32}) \right) n^{\left( b_{11}\kappa + b_{12}\kappa^{2} + b_{13} \right)x^{2} + (b_{21}\kappa + b_{22}\kappa^{2} + b_{23})x + b_{31}\kappa + b_{32}\kappa^{2} + b_{33}},$$
(13)

where  $x = \cos \Delta$ . x is a derived variable that optimizes the equation structure.





Again, these parameters are used as the inputs for the final multivariate least squares optimisation based on the Levenberg-Marquardt algorithm using all the simulation results. The final optimised values of the parameters in Eq. 13 are shown in Table 4. These values are similar to the initial parameter values obtained by the step-wise fitting process described above but they have been refined by global optimisation. The correlation coefficient between the prediction and simulation values based on the initial parameters (Table 3) is only 0.43 due to the error propagation in the step-wise fitting process. After global optimisation, the correlation coefficient increases significantly to nearly 1 (0.99) based on the values listed in Table 4.

Table 4: Parameters of the percolation equation in terms of fracture properties.

| a <sub>11</sub> | $a_{12}$        | $a_{21}$ | $a_{22}$ | a <sub>31</sub> | $a_{32}$        | $b_{11}$ | $b_{12}$ |
|-----------------|-----------------|----------|----------|-----------------|-----------------|----------|----------|
| 0.0643          | -0.1587         | -0.1188  | -0.5551  | 0.0549          | 0.9823          | -0.1612  | 0.2501   |
| $b_{13}$        | b <sub>21</sub> | $b_{22}$ | $b_{23}$ | b <sub>31</sub> | b <sub>32</sub> | $b_{33}$ |          |
| -0.0945         | 0.2633          | -0.0657  | 0.1313   | -0.2025         | 0.1519          | -0.4730  |          |

To visualize the relationships in Eq. 26, several surfaces of  $\bar{L}_t$  vs  $(n, \Delta)$  corresponding to different values  $\kappa$  (4, 18, 42 and 75) are shown in Figure 12. In general, higher  $\kappa$  values correspond to higher percolation threshold values. This is because higher  $\kappa$  values correspond to lower variation of fracture orientations, which leads to lower probabilities of fracture intersections. Consequently, this reduces the connectivity of the fracture network and hence longer fractures are needed to reach percolation.

Figure 12: Extracted surfaces from the final percolation threshold equation.



Eq. 13 is derived for the region of a dimensionless unit square, the result is expected to be applicable to areas at different scales  $(y \times y)$ . For these cases, the scaled percolation threshold  $\bar{L}_{ts}$  will be used (Eq. 14) revised from Eq. 13. If the average fracture length of a fracture network  $\bar{L} \geq \bar{L}_{ts}$ , there is more probability reach percolation.

$$\bar{L}_{ts} = \bar{L}_t \times y,\tag{14}$$

#### 3.3 Design of experiments for DFN with lognormal fracture lengths

Unlike the previous example, the fracture lengths in this section are modeled with a lognormal distribution. The mean fracture length  $\bar{L}$  and standard deviation  $\nu$ , which quantifies the dispersion of fracture lengths under the lognormal distribution, are used to derive the distribution parameters  $\mu$  and  $\sigma$ . These parameters, along with the probability density function, are calculated according to Eqs. 15-17. Five independent variables  $(\bar{L}, \nu, \Delta, \kappa)$  are considered with the aim of establishing the relationship  $n_t = f(\bar{L}, \nu, \Delta, \kappa)$ . In this context,  $n_t$  represents the fracture number threshold at which the percolation threshold may be

reached at a low probability (p0) in DFNs characterized by the parameters ( $\bar{L}$ ,  $\nu$ ,  $\Delta$ ,  $\kappa$ ). In Section 3.1, the exponential distribution of fracture length is defined by a single parameter. Consequently, the fracture length is selected to determine the threshold corresponding to p0. Given that the fracture length is governed by two parameters ( $\bar{L}$ ,  $\nu$ ), the parameters corresponding to the fracture length distribution are not selected. Instead, the fracture number is chosen.

$$\mu = \log(\bar{L}^2/\sqrt{\nu + \bar{L}^2}),\tag{15}$$

$$\sigma = \sqrt{\log(\nu / \bar{L}^2 + 1)},\tag{16}$$

$$f(\bar{L}|\mu,\sigma) = \frac{1}{\bar{L}\sigma\sqrt{2\pi}} e^{\frac{-(\ln\bar{L}-\mu)^2}{2\sigma^2}}, \bar{L} > 0, \tag{17}$$

Simulations and analyses were conducted corresponding to the 8×6×6×18×13 (684,400) variable combinations for the DFNs, with each case independently simulated 20 times. The number of variations for each variable are listed in Table 5.

Table 5: Parameters of DFNs in Section 3.3.


| Parameter             | Values analysed                               | Number of values |
|-----------------------|-----------------------------------------------|------------------|
| $\overline{n}$        | $round(10^i), i = 1.5, 1.65, 1.8, \dots, 2.4$ | 8                |
| $\overline{L}$        | 0.05,0.12,0.19,0.26,0.33,0.4                  | 6                |
| ν                     | 2,4,6,8,10,12                                 | 6                |
| $\Delta =  \mu - 90 $ | $(i-1) \times 5^{\circ}, i = 1,2,3,,18$       | 18               |
| κ                     | $round(10^i), i = 0.602,, 2$                  | 13               |

# 3.4 Derivation of percolation threshold equation for DFN with lognormal fracture lengths

The multivariable fitting process for the DFN with lognormal fracture lengths was analogous to that described in Section 3.2. Initially, the hypothetical relationship between  $n_t$  and  $\bar{L}$  was fitted (Eq. 18). Subsequently, by analysing the relationship between the parameters in the hypothetical model, new variables  $\nu$ ,  $\Delta$ , and  $\kappa$  were sequentially incorporated. The expressions for the newly added variables were then used to replace the parameters in the hypothetical model. Ultimately, this yields the fitted relationship between  $n_t$  and  $(\bar{L}, \nu, \Delta, \kappa)$ , with the fitting process detailed in Eqs. 18-21, where  $\kappa = \cos \Delta$ .

$$n_t = f(\bar{L}) = \alpha \bar{L}^b + c, \tag{18}$$

$$n_t = f(\bar{L}, \nu) = a_1 e^{a_2 \nu} \bar{L}^{b_1} + c_1 \nu^{c_2}, \tag{19}$$

$$n_t = f(\bar{L}, \nu, \Delta) = (a_{11}x + a_{12})e^{a_{22}\nu}\bar{L}^{b_{11}x + b_{12}} + c_{11}\nu^{c_{12}x + c_{13}},$$
(20)

$$n_{t} = f(\bar{L}, \nu, \Delta, \kappa) = \left[ (d_{1}\kappa^{2} + d_{2}\kappa + d_{3})x + (d_{4}\kappa^{2} + d_{5}\kappa + d_{6}) \right] e^{\left(d_{7}\frac{\kappa}{\bar{L}} + d_{8}\right)\kappa\nu} \bar{L}^{d_{9}x_{+}^{2}d_{10}\kappa + d_{11}} + d_{12}\nu^{d_{13}x^{2} + d_{14}},$$
(21)

Similarly, the parameters are used as inputs for the final multivariable least squares optimisation based on the Levenberg420 Marquardt algorithm, utilising all simulation results. The final optimised values of the parameters in Eq. 21 are presented in Table 6.

Table 6: Parameters of the percolation equation in terms of fracture properties.

| $d_1$   | $d_2$  | $d_3$    | $d_4$    | $d_5$    | $d_6$    | $d_7$    |
|---------|--------|----------|----------|----------|----------|----------|
| -0.0344 | 0.0560 | 0.2381   | 0.0335   | 0.0418   | 0.0053   | -0.0002  |
| $d_8$   | $d_9$  | $d_{10}$ | $d_{11}$ | $d_{12}$ | $d_{13}$ | $d_{14}$ |
|         |        |          |          |          |          |          |

To visualise the relationships in Eq. 21, several surfaces of  $n_t$  vs  $(\bar{L}, \nu, \Delta, \kappa)$  corresponding to different values of  $\nu$  (2, 4, and 12) and  $\kappa$  (2, 7, and 10) are presented in Figure 13. Generally, higher  $\Delta$  and lower  $\bar{L}$  result in increased percolation threshold  $n_t$ .

(c) 
$$v = 2, \kappa = 10$$
 (d)  $v = 12, \kappa = 10$ 

Figure 13: Extracted surfaces from the final percolation threshold equation.

Eq. 21 is derived for a dimensionless unit square, with its results expected to be applicable to regions of varying scales  $(y \times y)$ . For these scenarios, the percolation threshold  $\bar{n}_t$  will be adjusted using a scaling modification from Eq. 21, resulting in Eq. 22. If the number of fractures N is more than  $\bar{n}_{ts}$ , the probability of achieving percolation will be high.

$$n_{ts} = n_t \times y_n, \tag{22}$$

where  $y_n$  is the scaling correction factor,  $y_n = \log_{10} \left( \frac{1}{v} + 20 \right)$ .

## 440 4. Validation of the derived percolation threshold equation



## 4.1 Percolation analysis of fracture networks not used for deriving the threshold equation

(1) DFN with exponential fracture length. To test the performance of the derived zero percolation thresholds (Eq. 14), additional DFN models with different parameters ( $n = 100,200, \Delta = 3^{\circ}, 63^{\circ}, \kappa = 5,40,50$ ) at different scales ( $2m \times 2m$ ,  $60m \times 60m, 300m \times 300m, 900m \times 900m, 1100m \times 1100m, 1200m \times 1200m, 1300m \times 1300m$ ) were generated for percolation analysis. The percolation thresholds obtained from Eq. 14 and numerical simulation are shown in Figure 14. The close agreement between the predicted thresholds and the simulation results demonstrates that the derived relationships (Eq. 14) performed extremely well for predicting of zero percolation thresholds of DFNs with different parameters at different study scales.

Figure 14: Validation of Eq. 14 in study areas on different scales.

(2) DFN with lognormal fracture length. To test the derived zero percolation thresholds (Eq. 22), additional DFN models with different parameters ( $\bar{L} = 0.15, 0.25, \nu = 5, 7, \Delta = 26^{\circ}, 56^{\circ}, \kappa = 5, 9$ ) at different scales ( $2m \times 2m, 60m \times 60m, 300m \times 300m$ ) were generated for percolation analysis. The percolation thresholds obtained from Eq. 22 and numerical simulations were depicted in Figure 15. The remarkable concurrence between the predicted thresholds and the simulation outcomes underscores the robust performance of the derived relationships (Eq. 22) in accurately forecasting percolation thresholds across diverse parameter configurations and study scales for DFNs.



Figure 15: Validation of Eq. 22 in study areas on different scales.

## 4.2 Comparison with analytical solutions of simple fracture networks

The equations derived in Section 3.2 are for stochastic fracture networks with fracture lengths that follow an exponential distribution and fracture orientations that follow a von-Mises distribution. Because of the complexity of such fracture networks, there is no analytical solution for the corresponding percolation threshold. However, for simple fracture networks, where both fracture location and orientation follow completely random distributions and the fracture length is identical, the analytical solution for the percolation threshold of fracture length is (Balberg et al., 1984; Berkowitz, 1995):

$$L_C = 4.2/\sqrt{\pi\rho},\tag{23}$$

where  $\rho$  is the fracture density (=P20) calculated as  $\rho = n/y^2$  and  $y^2$  is the area of the study region. For the case of varying fracture length, the corresponding threshold is:

$$\bar{L}_t = \sqrt{L_C^2 - \sigma^2},\tag{24}$$

where  $\sigma^2$  is the variance of fracture length distribution. If the length follows an exponential distribution with parameter  $\lambda$ , Eq. 24 becomes (Berkowitz, 1995). Therefore, this section utilises fracture networks characterized by an exponential distribution of fracture lengths as a case study to compare the derived thresholds with analytical solutions.

$$\bar{L}_t = L_C/\sqrt{2} = 4.2/\sqrt{\frac{2\pi n}{y^2}} \approx 1.676 \ yn^{-0.5},$$
 (25)

This is a special case covered by the relationships derived in this work by setting  $\kappa = 0$  for a completely random distribution of fracture orientation and ignoring  $\cos \Delta$  as it is now irrelevant. Eq. 14 then becomes:

$$\bar{L}_{ts} = y \, a_{32} n^{b_{33}},\tag{26}$$

where  $a_{32} = 0.9823$ ,  $b_{33} = -0.4730$  The equation can be further simplified to:

$$\bar{L}_{ts} = 0.9823yn^{-0.4730},\tag{27}$$

which should be compared to Eq. 25. Note that the difference between the two equations is due to the different probabilities used to derive the percolation threshold. The theoretical solution is for a percolation probability of 50% while the derived relationship is for a percolation probability of 0%, as discussed above, and therefore it should be smaller.

There is a striking similarity between the analytical solution for p50 and the solution we derived for p0. The reasons why these two equations are so similar and what the factor of two in the first coefficient represents are important. These will be discussed in future work instead of here since it is not the focus of this work.

To compare the solutions, fracture networks with n = 100, 200, 300, 400 and 500 in a square of  $75m \times 75m$  are used for the simulations. Percolation thresholds corresponding to p0, p50 and p100 are calculated by Monte Carlo simulations and the results are shown in Figure 16. As demonstrated, the analytical solution is close to the p50 percolation threshold with an average absolute difference of 4.9%. On the other hand, the solution based on the derived equation is close to that of p0 with an average absolute difference of 11.4%.




Figure 16: Comparison of percolation thresholds determined by the analytical solution, the derived equation, and numerical

simulations. The points are the averages of 20 groups of MC simulations and the error bars are three times the corresponding standard deviations.

#### 4.3 Percolation analysis of real fracture networks using the derived equation

Two real fracture networks, as shown in Figure 17a and Figure 18a, are used to demonstrate further the application of the derived percolation threshold equations. Figure 17a shows a set of fractures traces on a rock outcrop taken from Wilson (Wilson, 2001). Figure 18a are fracture traces in the deformation bands on the Valley of Fire State Park, Nevada (Barton and Hsieh, 1989). Mid-points of the fractures are used to represent the fracture locations, as shown in Figure 17b and Figure 18b, respectively. They are all considered to follow approximately the Poisson distribution. The number of fractures, n, in the three systems are 35, 186. Clearly there is one dominant direction of fracture orientations in these systems, as illustrated in the rose diagrams shown in Figure 17c and Figure 18c. The orientation dispersion parameters ( $\kappa$ ) were calculated to be 145.18 and 25.68. For fracture length, the histogram in Figure 17d indicates an approximately exponential distribution for the first fracture set. For the second and third sets, the histograms (Figure 18d) suggest lognormal distributions. The average fracture lengths  $\bar{L}$  are 0.249m, 0.282mm, 0.9544m and the side lengths  $\gamma$  of study areas are 2.5m, 8mm, 7m, respectively.

Using Eqs. 21 and 22 with the parameters  $(\bar{L}, \nu, \Delta, \kappa)$  listed in Table 7, the calculated percolation thresholds  $(y_n \times n_t)$  in the horizontal and vertical directions can be calculated. For Figure 17a, the calculated percolation thresholds are 126 (horizontal) and 94 (vertical). The threshold in the horizontal direction is much greater than that in the vertical direction due to the fact that fractures are mainly vertical in this case. The fracture numbers are all less than these two thresholds, hence the fracture network is not percolated in both directions. This conclusion can easily be confirmed in this case by visual inspection of the fracture system displayed in Figure 17a.





(a) Fracture network from an outcrop

(b) Fracture centres

Figure 17: A set of fractures from an outcrop (Wilson, 2001) and the network properties.


Table 7: Parameters of three real fracture networks in Figure 17a and percolation assessment.

| Direction  | N  | Δ(°)  | v    | κ      | $\overline{L}$ | у    | $y_n \times n_t$ | $N \ge y_n \times n_t$ | Percolated |
|------------|----|-------|------|--------|----------------|------|------------------|------------------------|------------|
| Horizontal | 35 | 72.47 | 0.18 | 145.18 | 0.25m          | 2.5m | 126              | No                     | No         |
| Vertical   | 35 | 17.53 | 0.18 | 145.18 | 0.25m          | 2.5m | 94               | No                     | No         |

For the fracture set (Figure 18a), the average fracture length is 1.19m. The horizontal percolation threshold  $y \times \overline{L}_t$  is 1.34m and the vertical threshold is 0.69m. The average fracture length in this case is greater than the vertical threshold and therefore the fracture network is percolated vertically but not horizontally. On close inspection of Figure 18a, there is a cluster of fractures (marked in red) connecting the top and bottom sides of the study region.

Table 8: Parameters of three real fracture networks in Figure 18a and percolation assessment.

| Direction  | N   | Δ(°)  | κ     | у          | $\overline{L}$ | $y \times \overline{L}_t$ | $\bar{L} \geq y \times \bar{L}_t$ | Percolated |
|------------|-----|-------|-------|------------|----------------|---------------------------|-----------------------------------|------------|
| Horizontal | 146 | 80.54 | 25.68 | 7m         | 1.19m          | 1.34m                     | No                                | No         |
| Vertical   | 146 | 9.46  | 25.68 | 7 <i>m</i> | 1.19m          | 0.69m                     | Yes                               | Yes        |

Figure 18: Fracture traces of deformation bands in the Valley of Fire State Park, Nevada (Barton and Hsieh, 1989) and the corresponding network properties.

#### 5 Discussions



While the simplified relationships proposed in this study provide an efficient means of estimating the zero-percolation threshold directly from fracture network parameters, several limitations should be acknowledged. First, the present formulation is restricted to a two-dimensional (2D) domain with a single fracture set of similar orientation. This abstraction neglects the inherently three-dimensional (3D) nature of natural fracture systems, which typically comprise multiple intersecting sets formed under polyphase tectonic regimes. Extending the methodology to accommodate 3D configurations with multiple fracture sets will be a critical next step, although the associated analytical derivations are considerably more complex.

Second, the current approach relies on specific statistical assumptions for fracture attributes (e.g., Poisson-distributed locations, exponential or lognormal length distributions, and von Mises orientation distributions). While these choices are consistent with many prior DFN studies, natural fracture systems often exhibit spatial correlations, clustering, or heavy-tailed

distributions that may deviate from these idealizations. Future work will explore non-parametric estimation methods to mitigate dependency on pre-defined distribution forms and to better capture geological variability.

Third, the present formulation addresses only the geometric aspects of network connectivity. In real-world applications, hydraulic connectivity depends on both geometry and fracture transmissivity, which is influenced by aperture, roughness, and in-situ stress. Incorporating coupled flow–mechanical models would enable the framework to more directly predict hydraulic properties such as permeability.

A particularly promising avenue lies in integrating the derived functional forms of percolation thresholds into physics-informed machine learning workflows. The explicit parameter, threshold relationships, established here can serve as strong physical constraints within emerging architectures, such as Kolmogorov–Arnold Networks (KAN), to enhance generalizability and reduce training data requirements. Embedding these analytical insights into AI frameworks is expected to significantly improve predictive efficiency for complex, data-limited geological systems.

While the proposed 2D, single-set formulation constitutes a foundational step, its extension to multi-set, 3D, and distribution-flexible fracture systems, together with integration into advanced machine learning paradigms, offers a clear and impactful trajectory for future research.

## **6 Conclusions**







Percolation analysis of fracture networks is important for many applications, including oil and gas recovery, geothermal energy exploitation, hydrology, and groundwater protection in radioactive waste storage. In this paper, we focus on the percolation threshold relevant to rock impermeability, which is critically important for the safe underground storage of waste and energy materials.

Our approach to the calculation of the percolation threshold makes direct use of the characteristic parameters of 2D fracture network, in particular the number of fractures n, the fracture size (length)  $\bar{L}$  and the fracture orientation  $\Delta$ . This differs from the simplified approaches of using indirect characteristic parameters (e.g., fractal dimension), which could produce misleading results because fracture orientation is not considered. The assessment of fracture networks in this research was made under the following assumptions: (1). the centre points of fractures are randomly and independently distributed in space; (2). the lengths of fractures follow an exponential distribution; and (3). the orientations of fractures follow a von-Mises distribution, the parameters of which are the mean orientation  $\mu$  and the concentration parameter  $\kappa$ . The relationship between fracture network parameters and the corresponding percolation threshold is obtained from a large number of simulations. A non-linear multivariate fitting process was used to derive the final prediction equation for the percolation threshold in the form of  $\bar{L}_t = f(n, \cos\Delta, \kappa)$ . The derived equation provides a reliable relationship and an efficient way to estimate the connectivity and percolation state of a fracture network based directly on its parameters. The relationship was cross-validated using a published analytical solution and was further applied to three real fracture networks. The results demonstrate that the derived relationship can be used for fracture networks at different scales using a rescaling coefficient and can also be used for the

assessment of percolation in different directions. The derived relationship is a useful extension for rock impermeability evaluation (zero probability percolation p0), compared with the commonly used percolation assessment based on excluded volume, which corresponds only to the occurrence of percolation on average (i.e., 50% probability percolation, p50). Additionally, this work also studies fracture network models with log-normally distributed fracture lengths and derives zero percolation formulas, reaching conclusions similar to those mentioned above.

Owing to the complexity of multiple fracture sets, the present study is confined to a 2D fracture network with a single set. Future work will extend the framework to both 2D and 3D systems with multiple fracture sets, incorporating machine learning techniques to address the increased complexity.

# 7 Description of parameters



The parameter descriptions are shown in the table 9.

**Table 9: Description of parameters** 

| Parameter        | Explanation                                                                                  | Parameter     | Explanation                                                                         |
|------------------|----------------------------------------------------------------------------------------------|---------------|-------------------------------------------------------------------------------------|
| p0, p50,<br>p100 | Percolation thresholds of DFNs corresponding to 0%, 50%, and 100% percolation, respectively. | P30, P32      | Number and total fracture area of fractures per unit volume.                        |
| n                | Number of fractures in the network.                                                          | $\phi$        | Orientation angle of a fracture (measured from a reference direction, e.g., North). |
| $n_t$            | Fracture number threshold for p0.                                                            | μ             | Mean direction of fractures in the von-Mises distribution (primary orientation).    |
| L                | Length of a fracture.                                                                        | $\Delta$      | Relative orientation angle= $\ \mu - 90^{\circ}\ $ .                                |
| λ                | Exponential distribution parameter whose reciprocal represents mean fracture length.         | x             | $x = \cos\Delta$ .                                                                  |
| $ar{L}$          | Mean fracture length in DFN.                                                                 | К             | von-Mises concentration parameters, quantifying fracture orientation aggregation.   |
| σ                | Standard deviation of lognormal fracture lengths.                                            | у             | Scale of the study region.                                                          |
| v                | Variance of lognormal fracture lengths.                                                      | $y_n$         | Scaling correction factor.                                                          |
| $L_t$            | Critical fracture length threshold for p0.                                                   | $I_0(\kappa)$ | Modified Bessel function of order 0.                                                |
| P20, P21         | Fracture number and length per unit area.                                                    | ho            | Equal to P20                                                                        |

#### 8 Data availability

All the numerical simulation results are open-sourced and available at the Mendeley Data doi: 10.17632/y7yw25brph.1.

#### 9 Author contribution

Shaoqun Dong led the conceptualization, methodology development, and formal analysis of the study, drafted the original manuscript, and supervised the project. Lianbo Zeng contributed to methodology validation, data analysis, and manuscript revision. Chaoshui Xu provided critical methodological insights and analytical support. Peter Dowd contributed to theoretical discussions and manuscript refinement. Guohao Xiong assisted in numerical simulations and data visualization. Tao Wang supported geological interpretation and resource validation. Wenya Lyu participated in fracture network modeling and result visualization.

#### 10 Competing interests

The contact author has declared that none of the authors has any competing interests.

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
