# Peer review of "A Simplified Relationship Between the Zero-percolation Threshold and Fracture Set Properties"

_EGUsphere, 2025_

## Author Comment (AC3)

**Response to Reviewers' Comments**

We thank both reviewers and editors for their valuable and constructive comments and suggestions which have helped to improve the quality of the manuscript significantly. The manuscript has been revised extensively to improve the clarity of the technical content. The revised part is marked in red font. Revised Specific responses to reviewers' comments and suggestions are listed below.

**Response to Reviewer #1**

**1-1 Comments:** Several variables and equations are introduced early (e.g., $\mu$, $\kappa$, $\Delta$, $p_o$, $\overline{L}_t$) without immediate intuitive explanation. Consider including a summarized table of symbols and parameters, especially for readers unfamiliar with von Mises distribution or DFN modeling conventions.

**Response to 1-1:** Thank you sincerely for your valuable comments. In the revised manuscript, we have added a table (Table 9) providing the explanations of the symbols. Please refer to Table 9 on Line 590.

**1-2 Comments:** Machine learning offers a promising approach for capturing the complex relationship between fracture network properties and percolation behavior. The analytical formulation proposed in this work provides a strong foundation for integrating such physical insights into physics-informed machine learning frameworks. It is therefore recommended that the authors include a brief discussion on potential future directions, particularly the combination of their derived equations with emerging machine learning techniques, such as Kolmogorov–Arnold Networks (KAN), to enhance model generalizability and predictive capability.

**Response to 1-2:** Thank you very much for your valuable comments. A new Discussion section has been added. Limitations of this work and potential future directions have been added. Please refer to Lines 540-560.

**1-3 Comments:** Please ensure all citations are formatted uniformly (e.g., "Yi, Taverghi, 2009" vs. "Yi & Taverghi, 2009")

**Response to 1-3:** Thank you sincerely for your comments. We have thoroughly reviewed the entire manuscript and revised all citations to ensure uniform formatting. Please refer to Lines 30, 35, 40, 45, 50, 60, 65, 70, 75, 90, 110, 130 ,145, 150, 180, 190, 280, 520 and 535.

**1-4 Comments:** Equations are referenced inconsistently. A consistent format throughout will aid readability.

**Response to 1-4:** Thank you very much for your comments. We have carefully reviewed the entire manuscript and revised all equation references to ensure a consistent format throughout. Please refer to Lines 265, 270, 280, 285, 290, 300, 320, 330, 340, 350, 355, 360, 365, 375, 385, 390, 395, 400, 410, 415, 420, 425, 435, 440, 445, 450, 455, 460, 465, 470, 475, 480 and 510.

**1-5 Comments:** While the paper effectively presents a simplified relationship, a brief yet dedicated discussion on the inherent limitations of this simplification would further enhance the manuscript's academic rigor. Furthermore, outlining potential avenues for future research that could expand upon this simplified framework would significantly add to the paper's impact and forward-looking perspective.

**Response to 1-5:** Many thanks for your insightful comments. A discussion section has been added to address the inherent limitations of the simplified relationship and to outline potential future research directions. Please refer to Lines 540-560.

**Response to Reviewer #2**

**2-1 Comments:** The term "zero-percolation threshold" (p0) is introduced as a critical threshold for impermeability. How does p0 differ from p50 in real-world scenarios? Discuss how engineers should select p0 versus p50 thresholds for different applications and the trade-offs involved.

**Response to 2-1:** We greatly appreciate your valuable comments. A detailed explanation has been incorporated. Please refer to Lines 185-195.

**2-2 Comments:** The caption for Figure 5 states, "Each point is obtained by 400 times simulations, and in other words, one pair of (n, L, P) can be obtained via 20 MC simulations, and each of MC simulation is repeated 20 times and then averaged." This is confusing. Please rephrase to clarify the simulation process. For example, was a set of 20 DFN realizations generated to calculate a single probability P, and was this entire process repeated 20 times and averaged to get the

final plotted point? A clearer description of the simulation hierarchy is needed here and in the caption for Figure 6a.

**Response to 2-2:** Thank you sincerely for your valuable comments. The captions for Figures 5 and 6a, as well as the corresponding descriptions in the main text, have been revised to clearly explain the simulation hierarchy and process. Please refer to Lines 205-215, and 220.

**2-3 Comments:** The study establishes a relationship for a threshold fracture length, Lt. When discussing the exponential distribution, the text indicates that the average length is Lbar=1/λ. It is implied, but not explicitly stated, that Lt is this average length. This should be made explicit. More importantly, this definition is missing for the lognormal distribution (Sections 3.3-3.4). Please clarify what property of the lognormal distribution Lt represents (e.g., the mean, median, etc.).

**Response to 2-3:** Many thanks for your insightful comments. The description has been revised to define $\mu$ as the mean of the natural logarithm of fracture lengths and $\sigma$ as its standard deviation. It is now explicitly stated that $Lt$ equals the mean length $1/\lambda$ for the exponential distribution, and the median fracture length $e^{\mu}$ for the lognormal distribution. Please refer to Lines 230, 240 and 285.

**2-4 Comments:** The study commendably simplifies the problem by focusing on a single set of fractures. In the conclusion or discussion section, it would be beneficial to add a brief comment on the limitations of this assumption and how the proposed relationship might be extended in future work to handle DFNs with multiple fracture sets, which are common in real-world geological settings.

**Response to 2-4:** Thank you sincerely for your valuable comments. A new discussion section has been added to address the limitations of the current assumption of a single fracture set. This section also outlines potential extensions of the proposed relationship to accommodate DFNs with multiple fracture sets, which are prevalent in real-world geological settings. Please refer to Lines 540-585.

**2-5 Comments:** The manuscript is generally well-written, but there are minor issues with grammar and tense. For example, in the last sentence of the introduction, "The verification of the derived equations for zero-percolation will undergo a comprehensive series of tests in Section 4" should be in the past tense

(e.g., "was verified" or "is verified") as the paper is reporting completed work. A thorough proofread to correct such minor issues is recommended.

**Response to 2-5:** We appreciate your valuable comments. The manuscript has been systematically revised for grammar and tense, including the correction of the introduction sentence from "will undergo" to the past tense "underwent" (Line 121). All descriptions of completed derivations and experimental analyses have been adjusted to the past tense. Please refer to Lines 15, 20, 120, 250, 305, 310, 315, 410 and 455.